# Psychological Characteristics Associated with Post-Treatment Physical Status and Quality of Life in Patients with Brain Tumor Undergoing Radiotherapy

**DOI:** 10.3390/jpm12111880

**Published:** 2022-11-09

**Authors:** Anna Pieczyńska, Agnieszka Pilarska, Krystyna Adamska, Ewa Zasadzka, Katarzyna Hojan

**Affiliations:** 1Department of Occupational Therapy, Poznan University of Medical Sciences, 61-781 Poznan, Poland; 2Department of Rehabilitation, Greater Poland Cancer Centre, 61-866 Poznan, Poland; 3Department of Radiotherapy, Greater Poland Cancer Centre, 61-866 Poznan, Poland; 4Department of Elektroradiology, Poznan University of Medical Sciences, 61-701 Poznan, Poland

**Keywords:** radiation, cancer, brain tumor, personality, oncology, precision medicine

## Abstract

Radiotherapy (RT) is a mainstay of treatment for brain tumors. To minimize the risk of side effects while maximizing the therapeutic effects, personalized treatment plans, consisting mainly of genomics, radiomics, and mathematical modeling, are increasingly being used. We hypothesize that personality characteristics could influence treatment outcomes and thus could be used to help personalize RT. Therefore, the aim of this study was to identify the psychological characteristics associated with post-treatment physical status and quality of life (QoL) in patients with brain tumors undergoing RT. Two psychological tests—the Eysenck Personality Questionnaire and the State-Trait Anxiety Inventory—were administered prior to RT. Physical parameters before and after RT were also assessed through the following tests: hand grip strength, Timed Up and Go test, 6 Min Walk Test, and Functional Independence Measure. The Functional Assessment of Cancer Therapy–General (FACT-G) was used to assess QoL. The Functional Assessment of Chronic Illness Therapy–Fatigue (FACIT-F) was administered to assess fatigue. Neuroticism was significantly associated with low FACT-G Physical Well-Being scores. Psychoticism was associated with an improvement in physical fitness scores after RT. These findings suggest that personality traits should be considered when designing a personalized radiotherapy plan.

## 1. Introduction

In the year 2020, a total of 308,102 cases of brain tumor and other types of central nervous system (CNS) tumors were reported worldwide [1]. For most patients with brain tumors, radiotherapy (RT) is a mainstay of treatment [2,3]. However, treatment-related complications, which can negatively impact the quality of life (QoL), remain a major concern [4]. The likelihood of developing treatment-related adverse effects depends on various factors, including the total dose, the dose rate, the volume of healthy tissue irradiated, and the site of delivery [5]. The risk of adverse effects can be minimized while maximizing the therapeutic effects by designing a personalized approach to radiotherapy [6]. Recent molecular and imaging insights into cancer radiobiology are expected to provide a unique opportunity to develop patient-specific treatments, enabling the parallel design of next-generation trials to examine the effects adding targeted drugs to radiation; these trials will also allow for the critically important assessment of radiation volume and dose-limiting treatment toxicities [7]. In patients with primary malignant brain tumors, such as glioblastoma, standard RT plans can be improved by incorporating information from computational tumor models. These models, based on data from the patient’s medical scans, provide estimates of tumor infiltration, thus adding to the data obtained by conventional medical imaging to allow for personalized RT plans [5,8].

Fatigue is a common complication of brain RT, with an incidence rate >50% [4]. Symptoms usually begin within 2 weeks of starting RT, peaking at around 6 weeks. While fatigue may persist for several months, in most cases, this symptom improves gradually over time [4]. The fatigue level depends on the patient’s level of physical fitness, psychological status, and the presence (or not) of comorbid disorders, such as depression [9]. The association between increased fatigue and lower levels of physical fitness in cancer survivors is well documented [10].

The stress associated with a cancer diagnosis can lead to maladaptive behaviors, aggravate symptoms, and weaken the body [11]. The unique psychological characteristics of the individual partially determine the patient’s physiological response [11]. Personality traits—defined as an inherited set of personal characteristics that refer to individual differences in characteristic patterns of thinking, feeling, and behavior that determine how an individual reacts and adjusts to different situations—play a key role in the process of coping with and adapting to the disease [12]. Individual differences in personality may predispose an individual to the development of mental disorders and may intensify somatic ailments [12].

According to Eysenck et al., the permanent personality characteristics of every human being can be classified into three main traits: extraversion–introversion, neuroticism, and psychoticism [13]. Personality traits, especially the degree of neuroticism, are associated with the level of perceived distress and emotional stability, which often correlate with physical symptoms and the presence of disease [14]. Neuroticism is a personality trait characterized by the presence of anxiety, depression, and hostility associated with emotional instability and high basal arousal [15,16]. Individuals with high levels of psychoticism are characterized by psychophysical fitness, a sense of pleasure, and self-concentration [13]. Patients with high levels of psychoticism have a greater ability to isolate themselves from the emotions associated with difficult situations (e.g., cancer diagnosis). They are less likely to overanalyze the situation and ruminate. Consequently, these individuals often present better psychophysical functioning when facing illness.

To date, relatively few studies have evaluated the relationship between personality traits and functioning in cancer patients, although two studies recently evaluated this association in patients with prostate cancer [17] and skin cancer [18]. In addition, Bunevicius evaluated the role of personality in patients with brain tumors [19], particularly the relationship between personality dimensions and emotional and cognitive health status. However, to our knowledge, no studies have been performed to date to assess the association between personality traits and physical functioning in patients with brain tumors undergoing RT.

Numerous studies have examined the influence of personality on cancer risk, therapy, and prognosis [12,14,17,20,21]. Nonetheless, this relationship is still not well understood and remains controversial [17]. In addition, the presence of certain personality traits could lead to the development of maladaptive or unfavorable biological mechanisms, which may promote tumor progression and indicate a worse prognosis [17,21]. Depression and anxiety are common in cancer patients. Some studies have found an association between anxiety and/or depression and fatigue [21,22,23,24], but no comprehensive analysis has been performed to assess the influence of individual personality traits (e.g., neuroticism or psychoticism) on fatigue in patients with brain tumors undergoing radiation treatment.

Many studies have evaluated the role of individual patient characteristics, such as stress, anxiety, depression, and subjective QoL, on disease outcomes [12,14,19,24,25,26]. Although many studies have also evaluated personality traits (mainly neuroticism) [14,20,27,28,29], none of the clinical studies conducted to date have analyzed the association between personality traits and physical performance of patients with brain tumors treated with RT.

Just as all people are unique, so too are the specific characteristics of every cancer. Consequently, the treatment parameters must necessarily be designed to suit the individual [30]. In addition, due to human variability, the side effects of treatment may also vary and thus affect QoL in different ways. Personalized radiation therapy refers to an individual approach based on genomics, radiomics, and mathematical modeling. The literature review carried out by De Courcy et al. [5] showed that RT treatment personalization should take into account the sex of the patient. In our study, we decided to check whether personality traits can affect how patients tolerate RT in terms of physical fitness and QoL. We hypothesized that differences in personality could be used to personalize RT plans.

Determination of the patient’s psychological profile before starting RT can help healthcare professionals to identify patients in whom RT may be more likely to negatively impact QoL and physical function. By identifying these patients before treatment, it may be possible to develop and implement interventions to prevent or minimize the adverse effects of RT on physical performance and QoL. Clinically, it would be highly valuable to identify the psychological factors that influence the process of adaptation to the disease. It would also be beneficial to determine whether certain personality traits are associated with the somatic condition, symptom severity, physical fitness, and/or level of fatigue in patients with brain tumors undergoing RT.

Given the limited data on the potential influence of individual personality traits on physical functioning in patients with brain tumors undergoing RT, the aim of the present study was to identify the psychological characteristics associated with post-treatment physical status and QoL in patients with primary brain tumors treated with radiotherapy.

## 2. Materials and Methods

### 2.1. Study Design

This was a prospective clinical study conducted at the Radiotherapy Department at the Greater Poland Cancer Center in Poznan, Poland, between October 2021 and July 2022. The Ethics Committee of the Poznan University of Medical Sciences in Poznan approved the study protocol (No. 703/18). Written informed consent was obtained from all participants. This study was registered at ClinicalTrials.gov (identifier: NCT05192447) and was created as a result of the research project No. 2020/37/B/NZ7/01122 supported by the National Science Center.

### 2.2. Participants

Patients who met the study inclusion criteria were invited to participate by a radiation oncologist at the radiotherapy department. All patients were informed of the voluntary and confidential nature of the study. Participants were free to withdraw at any time for any reason. The inclusion criteria for the study were as follows: age of 18–70 years, diagnosis of a group III or IV CNS tumor (according to the 2021 World Health Organization Classification of Tumors of the CNS), eligibility for RT, good general physical condition (score of 0–2 on the Eastern Cooperative Oncology Group (ECOG) fitness scale), and signed informed consent. The exclusion criteria were: more than two brain lesions, psychological or psychiatric illness under pharmacological treatment, presence of other neurological disorders (e.g., multiple sclerosis, Parkinson’s disease, meningitis), and/or significant clinical circulatory failure (New York Heart Association scale, stage III or IV). Considering the pilot nature of the study, the sample size was determined for the correlation test, with an assumed test power of 80% and *p* = 0.05. The effect size was established based on previous publications on the relationship of personality in other types of cancer and was set at 0.5. For such assumptions, the minimum sample size is 28 participants.

### 2.3. Radiotherapy Procedure

All patients underwent intensity-modulated radiotherapy (IMRT) using a conventional fractionation scheme (2 Gy per dose, total dose = 60 Gy) administered over a 30-day period.

### 2.4. Study Data Collection

Demographic-, clinical-, and treatment-related data (sex, age, type of tumor, chemotherapy, type of tumor resection) were extracted from the medical records. The day before the start of RT, a qualified neuropsychologist assessed personality and anxiety using the instruments described below. Physical fitness, QoL, and the level of fatigue were assessed by a physiotherapist one day before the start of RT. These same assessments were repeated 24 h after the completion of RT.

### 2.5. Measurements

All participants underwent psychological assessments, which included a personality test and measures to assess anxiety levels. Physical parameters were also evaluated, including muscle strength, functional mobility and capacity, and functional independence. QoL and fatigue were also assessed. The specific tests are described below.

#### 2.5.1. Hand Grip Strength (HGS)

The HGS tests provide information indicative of overall muscle strength, muscle mass, physical function, and health and nutritional status. The HGS is predictive of mortality and hospital length of stay [31]. HGS tests are commonly used to evaluate older populations and in neurology [32]. HGS is also used to assess physical performance in cancer patients [33,34].

In this study, HGS was measured using a Jamar hydraulic dynamometer in accordance with the recommendations of the American Society of Hand Therapists [35]. The test position was as follows: the patient was seated in a chair without a backrest or armrests, with the feet placed in parallel and resting on the floor, and the hip and knee joints flexed at right angles. The arms were adducted to the trunk, the elbow was flexed at a right angle, the forearm was in a neutral position, and the wrist was in extension. Participants were asked to maintain the hand grip for 6 s at maximum intensity. Three measurements of the dominant hand were performed with a 1 min break between the measurements. The highest value was used for the analysis.

#### 2.5.2. Timed Up and Go Test (TUG)

The TUG is a reliable, validated test for quantifying functional mobility. The TUG can be used to assess patients at risk of deterioration of health and as a measure of response to treatment aimed at improving function and QoL. The test was originally developed for research in older populations, but is also used in younger groups [36]. To perform the test, patients are asked to sit in a chair with their back resting against the back, with their hands on the armrests. They are then told to stand up and walk at a normal speed (not fast) for a distance of 3 m (marked with tape on the floor). The test consists of getting up from the chair, walking 3 m, turning, returning to the chair, and sitting down. The test time began with the word “go” and ended when the participant was seated [36]. The task was performed three times, and the mean of three measurements was used for the analysis.

#### 2.5.3. 6 Min Walk Test (6MWT)

Functional capacity was estimated with the 6MWT, which is commonly used in clinical trials to estimate aerobic capacity in cancer patients [37]. The test was carried out in accordance with the guidelines of the American Thoracic Society [38]. The test was performed in a straight, flat hospital corridor (40 m in length). A total distance of 30 m was divided into sections of 3 m each marked with a tape on the floor. The patients rested for a minimum of 10 min before starting the test. Pulse, oxygen saturation, blood pressure, and subjective fatigue were measured before and after the test. The patient’s task was to walk at a natural pace for 6 min. The parameter of interest in the present study was the total distance walked.

#### 2.5.4. The Functional Independence Measure (FIM)

The FIM is an 18-item scale recommended for use in patients with neurological illnesses [39,40,41]. This scale is designed to evaluate physical, psychological, and social function, and it assesses performance in six areas: self-care, continence, mobility, transfers, communication, and cognition. The scale assesses the patient’s degree of dependence on the help of others in everyday activities. This tool is used to assess a patient’s level of disability and changes in response to rehabilitation or medical intervention [42].

#### 2.5.5. Quality of Life

The Functional Assessment of Cancer Therapy–General (FACT-G) was used to assess QoL. The FACT-G is a 27-item scale divided into four primary QoL domains: Physical Well-Being (PWB), Social/Family Well-Being (SWB), Emotional Well-Being (EWB), and Functional Well-Being (FWB). The highest possible score of the complete FACT-G is 108 points, with higher scores indicating better QoL [43].

#### 2.5.6. Assessment of Fatigue

The Functional Assessment of Chronic Illness Therapy–Fatigue (FACIT-F) scale was used to assess fatigue symptoms [44]. This tool consists of 13 items to assess patient-reported fatigue over the last 7 days. Responses are given on a 5-point Likert scale and range from 0 to 4. Total scores range from 0 to 52, with higher scores indicating less fatigue. Scores < 30 are considered to indicate severe fatigue. We chose the FACIT-F as a performance measure because it is widely used in cancer fatigue studies and has good internal consistency and test–retest reliability [44].

#### 2.5.7. Personality Test

The short version of the Eysenck Personality Questionnaire (EPQ-R) was used to evaluate personality. The developers of the EPQ-R [13] created an abbreviated version—the EPQ-R (S)—for use in situations where time is limited. The questionnaire is designed for people 16 to 69 years. The EPQ-R (S) contains 48 questions with two response options (yes or no). The results are tabulated into four 12-point scales (Psychoticism, Extraversion, Neuroticism, and Lie) and the corresponding sten score. Responses considered “correct” receive 1 point, and those that are “incorrect” receive 0 point. The maximum score for each scale is 12 points (1 point for each item).

#### 2.5.8. Anxiety Level Assessment

The State-Trait Anxiety Inventory (STAI) [45] was used to test anxiety levels. The STAI was designed to study anxiety, understood as a transient and situational state of an individual, as well as anxiety as a relatively constant personality trait. The tool consists of two subscales. The first subscale (X-1) measures state anxiety, and the second one (X-2) measures trait anxiety. Each subscale consists of 20 items with four response options (1—not at all, 2—somewhat, 3—moderately so, 4—very much so). Point values can range from 20 to 80 points. The level of anxiety is indicated by the total number of points obtained, with higher scores indicating greater levels of anxiety. The developers of this tool define anxiety as a trait as a theoretical construct, denoting an acquired behavioral disposition that makes an individual susceptible to perceiving a wide range of objectively harmless situations as threatening and reacting to them with a disproportionately strong state of anxiety relative to the size of the objective danger. This definition emphasizes the learned nature of fear.

### 2.6. Statistical Analysis

Statistical analysis was performed with the Statistica 13.3 software (TIBCO Software, Poland). *p* < 0.05 was considered statistically significant. The Shapiro–Wilk test was used to check the distribution normality. In the absence of a normal distribution and for ordinal variables, the data are presented as a median and a range. The Wilcoxon signed-rank test was used to compare the parameters before and after RT. Correlations were assessed with the Spearman test.

## 3. Results

A total of 29 patients were enrolled in the study. Figure 1 shows the study flow diagram. Table 1 shows the general characteristics of the participants. The mean age was 53 years. Most participants (65%) were women. The diagnosis in most cases (89.7%) was GBM. Twelve patients (41.4%) underwent complete tumor resection. Most patients (79.3%) underwent chemotherapy. None of the patients presented acute encephalopathy after radiotherapy.

In terms of personality traits, most respondents had a moderate intensity for the four variables, with high scores most often related to the traits of neuroticism, low extraversion, and psychoticism (Figure 2). None of the respondents obtained a low result on the neuroticism subscale.

The distribution of anxiety level as a trait was close to normal (Figure 3), with more than one-third of the participants (37.8%) presenting a moderate level of anxiety. Very low and very high results were observed in a similar percentage of patients (6.9%). Several respondents reported feeling anxiety at a low and high level. In terms of state anxiety, none of the participants had a very low level of anxiety, and only four (13.8%) described it as low. Most of the participants had a high (37.9%), medium (27.6%), or very high level (20.7%) of state anxiety.

Table 2 compares the pre- and post-RT results for fitness, showing an increase in HGS, a shorter TUG test time, and a longer distance covered during the 6MWT. However, these differences are not statistically significant. There were also no significant differences in pre- and post-RT QoL. The level of fatigue (FACIT-F scores) increased significantly after RT (*p* = 0.04).

Table 3 shows the relationship between personality traits and the results of fitness tests, QoL, and the level of fatigue before RT. As the table shows, the more features of extraversion, the higher the score on the FACT-G, and thus the better the QoL. The more features of psychoticism, the better the results on the 6MWT, and the worse the results of the QoL in the social sphere. Neuroticism negatively correlated with the results on the FACT-PWB subscale.

This analysis shows an association between both trait and the patient-reported level of anxiety before RT treatment with functional independence. Higher levels of anxiety were associated with lower scores on the FIM scale (Table 4).

After the completion of RT, psychotic traits were positively correlated with the 6MWT test results and negatively correlated with FACT-PWB and neurotic traits. Extraversion and lying did not present significant correlations (Table 5).

As Table 6 shows, both trait and state anxiety were negatively correlated with the FIM scale. Stage anxiety was positively correlated with the TUG test results and negatively correlated with the FACT-G.

Table 7 shows the correlations between the pre- and post-RT differences with personality traits. Positive correlations were found between psychoticism and 6MWT and FACT-G in the domains of Physical and Social Well-Being, indicating that patients with more features of psychoticism improved more on the 6MWT after RT and also had a better QoL.

No significant correlations were found between the pre- and post-RT differences and the level of trait and state anxiety (Table 8).

## 4. Discussion

The main aim of this study was to identify the psychological characteristics associated with post-treatment physical status and QoL in patients with brain tumors undergoing RT. To our knowledge, this is the first study to investigate this type of relationship in patients with primary brain tumors receiving RT. The findings of this study show that personality traits are associated with physical fitness and subjective QoL before and after radiation therapy in patients with brain tumor.

In this study, we found a positive correlation between the trait of extraversion and QoL (general FACT-G score) before RT. This association in extroverts is most likely attributable to specific behaviors, perceptions, and certain emotions in these individuals. Extraversion is commonly associated with a high level of optimism, the capacity to easily establish relationships (and thus a higher probability of having social support), less anxiety, low hopelessness, less likelihood of feeling sad and anxious, good adaptation to new situations, and self-confidence and assertiveness [13]. Macia et al. [28] showed that a low level of neuroticism and a high level of extraversion appear to be protective factors for mental health in people with cancer. The same may be true of physical health. However, we found no association between objective measures of physical functioning, such as muscle strength, functional mobility and capacity, and personality traits. Nevertheless, when we analyzed the subjective feelings of patients, we observed that people with high levels of extraversion and low levels of neuroticism reported better functioning and QoL related to physical health (Table 3 and Table 5).

The association between neuroticism and low scores on the Physical Well-Being section of the FACT-G (which includes questions about feeling pain, feeling sick, or lack of energy) is most likely due to a tendency among neurotic patients to present high levels of anxiety with an excessive focus on the disease and a lack of self-confidence and to expect treatment failure (Table 3 and Table 5). Previous research has shown that people with a high level of neuroticism are more likely to believe that they have a greater risk of getting sick and of experiencing a worse course of the disease than other people [14]. Krok et al. [20] showed that increased neuroticism is associated with higher pain intensity and greater fatigue. According to Hoeger et al. [14], neuroticism may make patients more sensitive to the presence of negative affect. These findings are consistent with our results, in which high levels of neuroticism were associated with lower Physical Well-Being scores both before and after RT. Similarly, Dahl et al. [17] found that a high level of neuroticism (assessed before treatment) in patients with prostate cancer was significantly associated with a higher percentage of overall health problems both before and after treatment.

In our study, a high level of psychoticism was associated with a good result in 6MWT and a worse QoL especially in the social functioning (before RT). In previous research has shown that individuals with a high level of psychoticism are indifferent to other people and avoid close relationships [35]. Psychotic individuals show little sensitivity to external stimuli and do not show empathy towards others. These characteristics can be advantageous in certain difficult situations (e.g., in cancer treatment), as they reduce the risk of overstimulation and mental tension, which results in less fatigue and a better mood [22,25]. Spielberger et al. [45] showed a significant positive association between psychoticism and behavior in the form of increased activity, elevated mood, and lack of inhibitions.

We found that lower levels of anxiety were associated with a better pretreatment QoL in the social sphere. As previous studies have shown [23,26,46], the level of social support has a direct impact on the level of anxiety experienced in cancer patients. People who report little to no social support usually have more stress, anxiety, and lower QoL [23,26,46]. We also found that patients with a high index of psychoticism improved QoL (physical subscale) and also increased the distance on the 6MWT after RT when compared with pretreatment values. These findings can be explained by the high value that psychotic individuals give to psychophysical fitness. In addition, these individuals tend to suffer from less anxiety, report indifference to the environment, and have a low level of empathy. This may affect the task-oriented approach to the rehabilitation process, in that such individuals do not care about negative emotional reactions from the environment, and are interested in maintaining physical fitness. Earlier studies in cancer patients mainly focused on the relationship between psychoticism and mental functioning. Psychoticism has been shown to be a predictor of depressive symptoms in breast cancer survivors [47] and depressive symptomatology correlated with psychoticism in skin cancer patients [18]. Our study was the first to investigate the relationship between psychoticism and physical functioning in patients with brain tumors.

As regards the patients’ physical status before and after RT, the only statistically significant decrease in this study was in the level of fatigue. By contrast, strength and functional mobility and capacity slightly increased, while QoL was unchanged. These results are consistent with the findings of other authors, such as Bitterlich et al. [48]. They also observed a statistically significant increase in fatigue after RT but no significant differences in QoL. We did not assess long-term changes in QoL, although a decrease in QoL is one of the well-known late adverse events after RT [49].

## 5. Strength and Limitations

The sample size is a main limitation of this study. However, the size of the study group was similar to other studies in this clinical population (brain tumor patients undergoing radiotherapy) [47]. The number of patients included in the study was due to two factors: the pilot nature of the study and the characteristic of patients with brain tumors, who often did not want to participate in this clinical observation due to symptoms such as fatigue, anxiety, and malaise. Several patients also withdrew their consent during RT. In addition, due to the sample size, we were only able to perform basic statistical analyses. By contrast, the main strength of this study is that it is one of the first studies about the association between personality traits and physical performance of patients with brain tumors treated with RT.

## 6. Conclusions

The findings of this study show that personality traits (especially neuroticism and psychoticism) and anxiety levels are associated with physical fitness and subjective QoL before and after radiation therapy in patients with brain tumor. Our preliminary results suggest that patients with high levels of neuroticism may have a worse tolerance for RT in terms of physical functioning and QoL. By contrast, patients with high levels of psychoticism appear to present a lower risk of suffering a decline in physical functioning after RT treatment. These outcomes suggest that personality traits should be considered when developing personalized radiation therapy treatments. In particular, patients with brain tumors with certain personality types may require additional rehabilitation during RT treatment. Our results indicate the necessity of further research in the area of personality relationship with RT tolerance in patients with brain tumors. Subsequent studies should be on a larger study sample, and with a longer observation period, using statistical modeling, so that their results can be generalized to the entire population.

## Figures and Tables

**Figure 1 jpm-12-01880-f001:**
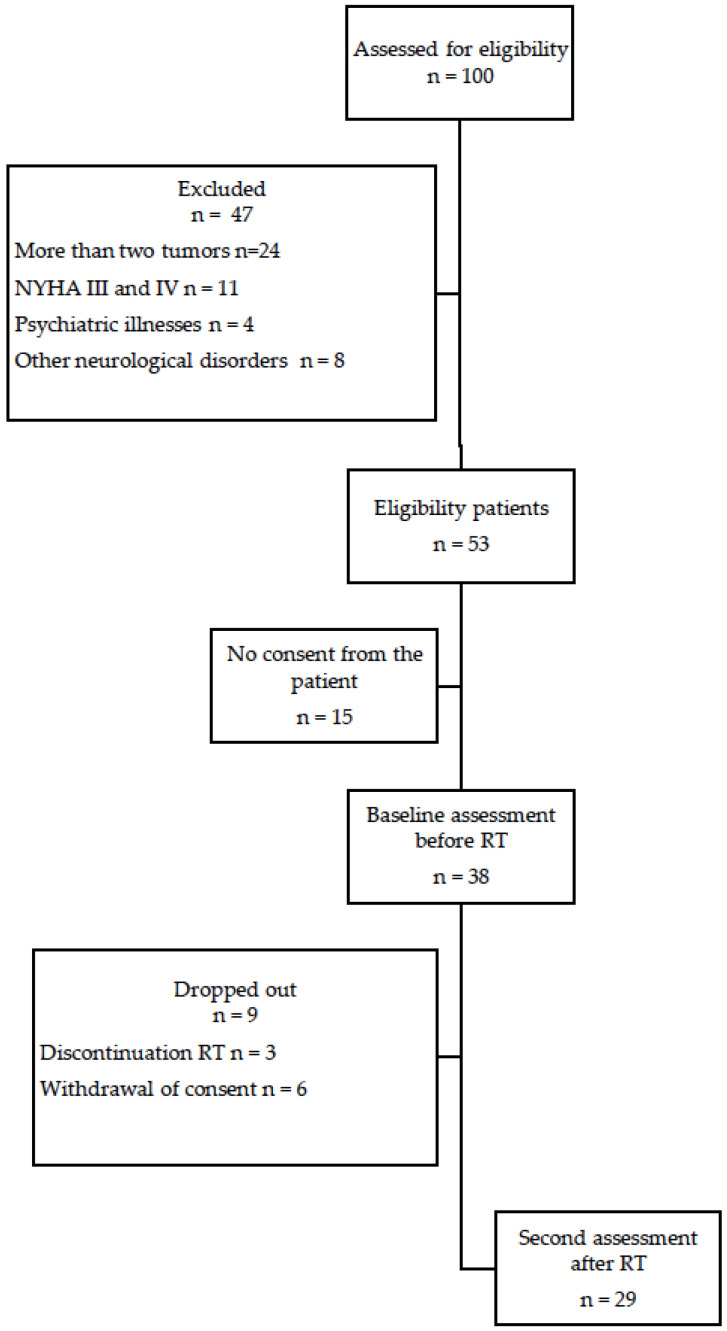
The flow of participants through the study.

**Figure 2 jpm-12-01880-f002:**
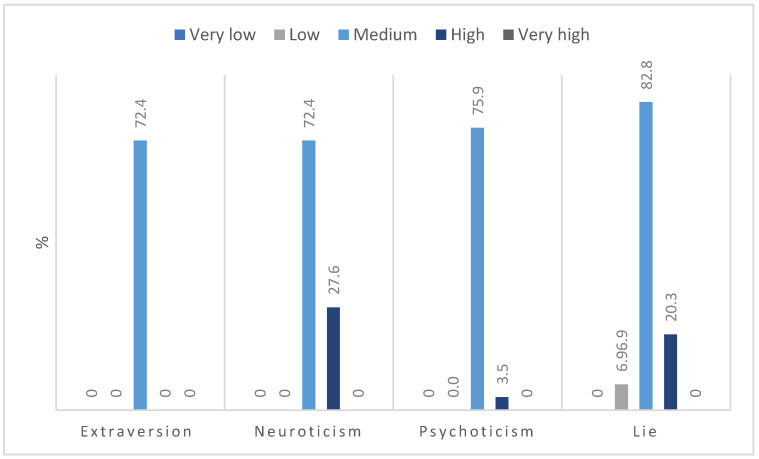
Intensity of personality traits.

**Figure 3 jpm-12-01880-f003:**
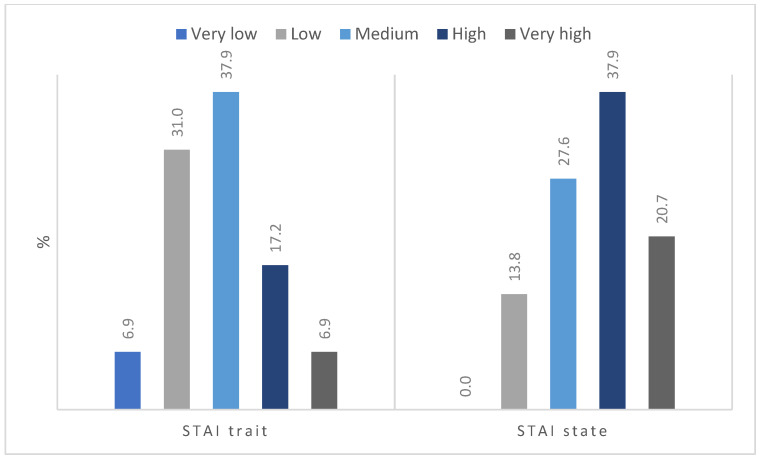
Intensity of anxiety level.

**Table 1 jpm-12-01880-t001:** Demographic and clinical characteristics of the study group.

Characteristic	Participants (n = 29) n (%) or Mean ± SD
Age, years	52.8 ± 14.1
Sex	
Female	10 (34.5)
Male	19 (65.5)
Type of tumor	
GBM	26 (89.7)
Oligodendroglioma	2 (6.9)
Ependymoma	1 (3.5)
Total resection	
Yes	12 (41.4)
No	17 (58.6)
Chemotherapy	
Yes	23 (79.3)
No	6 (20.7)

Abbreviations: SD, standard deviation; GBM, glioblastoma multiforme.

**Table 2 jpm-12-01880-t002:** Physical fitness, quality of life, and level of fatigue before and after radiotherapy.

Parameters Median (Range)	Before RT	After RT	*p*-Value
HGS (kg)	28 (11.3–53)	30 (13–4.5)	0.659
TUG (s)	8.38 (5.28–12.95)	8 (5.12–13.76)	0.331
6MWT (m)	405 (267–670)	431 (243–703)	0.666
FIM (points)	126 (7–126)	126 (5–126)	0.683
FACT-G PWB (points)	22 (10–28)	21 (10–28)	0.201
FACT-G SWB (points)	24 (9–28)	24 (2–28)	0.332
FACT-G EWB (points)	15 (0–24)	15 (0–24)	0.550
FACT-G FWB (points)	19 (2–28)	19 (4–28)	0.984
FACT-G total (points)	79.3 (43–105)	77.5 (43–98)	0.648
FACIT-F (points)	38 (11–52)	33 (14–50)	0.040

Abbreviations: RT—radiotherapy; HGS—hand grip strength; 6MWT—6 Min Walk Test; FIM—Functional Independence Measure; FACT-G—Functional Assessment of Cancer Therapy–General; PWB—Physical Well-Being; SWB—Social Well-Being; EWB—Emotional Well-Being; FWB—Functional Well-Being; FACIT-F—Functional Assessment of Chronic Illness Therapy–Fatigue.

**Table 3 jpm-12-01880-t003:** Correlation between personality traits and the results of fitness tests, quality of life, and fatigue levels before radiotherapy.

Test	Extraversion		Neuroticism		Psychoticism		Lie	
	r	*p*-Value	r	*p*-Value	r	*p*-Value	r	*p*-Value
HGS	0.049	0.81	0.253	0.19	0.307	0.11	−0.359	0.06
TUG	−0.192	0.33	0.145	0.46	−0.300	0.12	−0.010	0.96
6MWT	0.076	0.71	−0.167	0.42	0.402	0.04	−0.097	0.64
FIM	0.079	0.69	−0.246	0.21	0.023	0.91	−0.066	0.74
FACT-G PWB	0.087	0.66	−0.399	0.04	−0.117	0.55	0.373	0.05
FACT-G SWB	0.420	0.03	−0.309	0.11	−0.239	0.02	−0.088	0.66
FACT-G EWB	0.378	0.05	−0.186	0.34	−0.023	0.91	−0.033	0.87
FACT-G FWB	0.455	0.02	−0.265	0.17	−0.134	0.49	−0.124	0.53
FACT-G total	0.428	0.02	−0.335	0.08	−0.136	0.49	−0.010	0.96
FACIT-F	0.056	0.78	−0.159	0.42	−0.008	0.97	0.336	0.08

Abbreviations: RT—radiotherapy; HGS—hand grip strength; 6MWT—6 Min Walk Test; FIM—Functional Independence Measure; FACT-G—Functional Assessment of Cancer Therapy–General; PWB—Physical Well-Being; SWB—Social Well-Being; EWB—Emotional Well-Being; FWB—Functional Well-Being; FACIT-F—Functional Assessment of Chronic Illness Therapy–Fatigue; r—Spearman’s rank correlation.

**Table 4 jpm-12-01880-t004:** Correlation between anxiety level and fitness tests, quality of life, and level of fatigue before RT.

Measurements before RT	STAI Trait		STAI State	
	r	*p*-Value	r	*p*-Value
HGS	0.113	0.57	−0.084	0.67
TUG	0.236	0.23	0.36	0.06
6MWT	−0.233	0.25	−0.343	0.09
FIM	−0.407	0.03	−0.479	0.01
FACT-G PWB	−0.329	0.09	−0.371	0.05
FACT-G SWB	−0.379	0.05	−0.351	0.07
FACT-G EWB	−0.143	0.47	−0.171	0.38
FACT-G FWB	−0.189	0.33	−0.360	0.06
FACT-G total	−0.291	0.13	−0.386	0.04
FACIT-F	−0.267	0.17	−0.280	0.15

Abbreviations: RT—radiotherapy; HGS—hand grip strength; 6MWT—6 Min Walk Test; FIM—Functional Independence Measure; FACT-G—Functional Assessment of Cancer Therapy–General; PWB—Physical Well-Being; SWB—Social Well-Being; EWB—Emotional Well-Being; FWB—Functional Well-Being; FACIT-F—Functional Assessment of Chronic Illness Therapy–Fatigue; r—Spearman’s rank correlation.

**Table 5 jpm-12-01880-t005:** Correlations between personality traits and results of fitness tests, quality of life, and level of fatigue assessed after RT.

Measurements after RT	Extraversion		Neuroticism		Psychoticism		Lie	
	r	*p*-Value	r	*p*-Value	r	*p*-Value	r	*p*-Value
HGS	0.050	0.81	0.217	0.29	0.231	0.27	−0.212	0.31
TUG	0.045	0.83	0.135	0.51	−0.148	0.47	−0.153	0.46
6MWT	0.085	0.69	−0.226	0.28	0.561	0.00	−0.029	0.89
FIM	−0.040	0.85	−0.203	0.33	0.040	0.85	−0.181	0.39
FACT-G PWB	−0.017	0.93	−0.441	0.02	0.269	0.18	0.216	0.29
FACT-G SWB	0.194	0.34	−0.037	0.86	−0.014	0.95	0.266	0.19
FACT-G EWB	0.267	0.19	−0.198	0.33	−0.078	0.70	−0.128	0.53
FACT-G FWB	0.301	0.14	−0.309	0.13	0.124	0.55	−0.178	0.38
FACT-G total	0.304	0.13	−0.362	0.07	0.147	0.47	−0.026	0.90
FACIT-F	0.093	0.66	−0.087	0.68	0.277	0.18	0.257	0.22

Abbreviations: RT—radiotherapy; HGS—hand grip strength; 6MWT—6 Min Walk Test; FIM—Functional Independence Measure; FACT-G—Functional Assessment of Cancer Therapy–General; PWB—Physical Well-Being; SWB—Social Well-Being; EWB—Emotional Well-Being; FWB—Functional Well-Being; FACIT-F—Functional Assessment of Chronic Illness Therapy–Fatigue; r—Spearman’s rank correlation.

**Table 6 jpm-12-01880-t006:** Correlation between anxiety level and results of fitness tests, quality of life, and level of fatigue after RT.

Measurements after RT	STAI Trait		STAI State	
	r	*p*-Value	r	*p*-Value
HGS	0.038	0.86	−0.190	0.36
TUG	0.282	0.16	0.449	0.02
6MWT	−0.299	0.16	−0.346	0.09
FIM	−0.397	0.05	−0.506	0.01
FACT-G PWB	−0.359	0.07	−0.379	0.06
FACT-G SWB	−0.115	0.58	−0.092	0.66
FACT-G EWB	−0.065	0.75	−0.110	0.59
FACT-G FWB	−0.230	0.26	−0.490	0.01
FACT-G total	−0.301	0.14	−0.444	0.02
FACIT-F	−0.191	0.36	−0.149	0.48

RT—radiotherapy; HGS—hand grip strength; 6MWT—6 Min Walk Test; FIM—Functional Independence Measure; FACT-G—Functional Assessment of Cancer Therapy–General; PWB—Physical Well-Being; SWB—Social Well-Being; EWB—Emotional Well-Being; FWB—Functional Well-Being; FACIT-F—Functional Assessment of Chronic Illness Therapy–Fatigue; r—Spearman’s rank correlation.

**Table 7 jpm-12-01880-t007:** Correlations between differences in outcomes before and after RT and personality traits.

Difference in Measurements before and after RT	Extraversion		Neuroticism		Psychoticism		Lie	
	r	*p*-Value	r	*p*-Value	r	*p*-Value	r	*p*-Value
HGS	0.178	0.38	−0.227	0.27	−0.282	0.16	0.288	0.15
TUG	0.275	0.17	−0.056	0.79	0.001	0.99	−0.238	0.24
6MWT	0.059	0.78	−0.132	0.53	0.446	0.03	0.126	0.55
FIM	−0.031	0.89	−0.047	0.82	−0.159	0.44	−0.102	0.62
FACT-G PWB	−0.007	0.97	−0.009	0.97	0.399	0.04	−0.172	0.40
FACT-G SWB	−0.021	0.92	0.197	0.34	0.466	0.02	0.165	0.43
FACT-G EWB	−0.141	0.49	−0.070	0.73	−0.133	0.52	−0.019	0.94
FACT-G FWB	−0.196	0.34	−0.071	0.73	0.341	0.09	−0.031	0.88
FACT-G total	−0.156	0.45	−0.035	0.87	0.412	0.04	−0.093	0.65
FACIT-F	0.172	0.41	−0.062	0.79	0.311	0.13	−0.280	0.18

Abbreviations: RT—radiotherapy; HGS—hand grip strength; 6MWT—6 Min Walk Test; FIM—Functional Independence Measure; FACT-G—Functional Assessment of Cancer Therapy–General; PWB—Physical Well-Being; SWB—Social Well-Being; EWB—Emotional Well-Being; FWB—Functional Well-Being; FACIT-F—Functional Assessment of Chronic Illness Therapy–Fatigue; r—Spearman’s rank correlation.

**Table 8 jpm-12-01880-t008:** Correlations between differences in outcomes before and after RT and anxiety level.

Difference in Measurements before and after RT	STAI Trait		STAI State	
	r	*p*-Value	r	*p*-Value
HGS	−0.142	0.49	−0.195	0.34
TUG	0.060	0.77	0.132	0.52
6MWT	−0.102	0.63	−0.183	0.38
FIM	0.083	0.69	−0.042	0.84
FACT-G PWB	0.048	0.82	0.090	0.66
FACT-G SWB	0.082	0.69	0.054	0.79
FACT-G EWB	0.044	0.83	−0.041	0.84
FACT-G FWB	−0.154	0.45	−0.257	0.21
FACT-G total	−0.002	0.99	−0.117	0.57
FACIT-F	0.043	0.85	0.071	0.74

RT—radiotherapy; HGS—hand grip strength; 6MWT—6 Min Walk Test; FIM—Functional Independence Measure; FACT-G—Functional Assessment of Cancer Therapy–General; PWB—Physical Well-Being; SWB—Social Well-Being; EWB—Emotional Well-Being; FWB—Functional Well-Being; FACIT-F—Functional Assessment of Chronic Illness Therapy–Fatigue; r—Spearman’s rank correlation.

## Data Availability

The datasets generated for this study are available on request to the corresponding author.

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
