# Peer review of "Psychological Characteristics Associated with Post-Treatment Physical Status and Quality of Life in Patients with Brain Tumor Undergoing Radiotherapy"

_jpm, 2022, doi:10.3390/jpm12111880_

Round 1

Reviewer 1 Report

This manuscript was described about the psychological characteristics before and after radiation therapy for brain tumor. The authors had collected detailed data and this manuscript was well written. However, there are some problems in this manuscript.

 One was the small sample size.

 In case of brain tumor who undergo radiation therapy, mental status and QoL may change during radiation therapy due to the effects of tumor progression and brain edema. Therefore, authors should evaluate and discuss the conditions of brain such as the changes of tumor and brain edema in the pre- and post-treatment.

 Line 377-379: the finding that a high level of psychoticism was associated with a good 377 result in 6MWT and a worse QoL in the social sphere before RT can be explained by a 378 task-oriented approach to stress or challenge situations. Was this result due to the subject who had a brain tumor? The authors should discuss about other types of tumors.

Author Response

Dear Reviewer 1,

 We would like to express our gratitude for the Reviewers’ invaluable comments and suggestions for the improvements to our manuscript. All comments have been addressed and all changes suggested by the Reviewer have been implemented. Our answers are listed below.

This manuscript was described about the psychological characteristics before and after radiation therapy for brain tumor. The authors had collected detailed data and this manuscript was well written. However, there are some problems in this manuscript.

Comment 1. One was the small sample size.

Thank you very much for your comment. We agree that the sample size used in our study is not sufficient to generalize the results to the entire population. However, we noted in limitation that the study is a pilot study, which we additionally emphasized in the material and methods section. Earlier studies in the group of patients with brain tumors undergoing radiation therapy have similar sample sizes. We present a few of them below.

Kepka L, Tyc-Szczepaniak D, Osowiecka K, Sprawka A, TrÄ…bska-Kluch B, Czeremszynska B. Quality of life after whole brain radiotherapy compared with radiosurgery of the tumor bed: results from a randomized trial. Clin Transl Oncol. 2018;20(2):150-159. doi:10.1007/s12094-017-1703-5 

Bitterlich C, Vordermark D. Analysis of health-related quality of life in patients with brain tumors prior and subsequent to radiotherapy. Oncol Lett. 2017;14(2):1841-1846. doi:10.3892/ol.2017.6310

Kim KS, Wee CW, Seok JY, et al. Hippocampus-sparing radiotherapy using volumetric modulated arc therapy (VMAT) to the primary brain tumor: the result of dosimetric study and neurocognitive function assessment. Radiat Oncol. 2018;13(1):29. doi:10.1186/s13014-018-0975-4

In addition, the sample size was determined for the correlation test, with the assumed test power of 80% and p = 0.005. The effect size was established based on previous publications on the relationship of personality in other types of cancer and was set at 0.5. For such assumptions, the minimum sample size is 28 people.

We recognize that further multicentre studies, in larger groups and with longer follow-up times, are needed to generate more significant data. However, we are of the opinion that our pilot study will encourage other researchers to undertake further research on this important subject regarding personalized radiotherapy.

Comment 2. In case of brain tumor who undergo radiation therapy, mental status and QoL may change during radiation therapy due to the effects of tumor progression and brain edema. Therefore, authors should evaluate and discuss the conditions of brain such as the changes of tumor and brain edema in the pre- and post-treatment.

Thank you very much for this important observation. We fully agree that radiation complications are a major concern that can negatively impact a patient's quality of life. The follow-up examination after radiotherapy was carried out 24 hours after the end of the treatment, so we could only consider acute complications that occurred during the radiation or up to a month after the end of the treatment. They mainly include acute encephalopath and fatigue1. None of the patients included in the study presented acute encephalopath. This information was added in the "results" section. In turn, the level of fatigue is the subject of this study. We agree that longer-term studies should include an analysis of complications such as radionecrosis, pseudoprogression, and leukoencephalopathy. When collecting further data, we analyze the results of MR, which is performed at the earliest 3 months after the end of treatment.

Barisano G, Bergamaschi S, Acharya J, et al. Complications of Radiotherapy and Radiosurgery in the Brain and Spine. Neurographics (2011). 2018;8(3):167-187. doi:10.3174/ng.1700066

Comment 3. Line 377-379: the finding that a high level of psychoticism was associated with a good 377 result in 6MWT and a worse QoL in the social sphere before RT can be explained by a 378 task-oriented approach to stress or challenge situations. Was this result due to the subject who had a brain tumor? The authors should discuss about other types of tumors.

Thank you very much for pointing this inaccuracy. It is related to a mistake during the linguistic proofreading which we did not notice. The sentence has been changed.

Reviewer 2 Report

This study aimed to evaluate the psychological/personality factors associated with post-treatment physical status and quality of life (QOL) in patients with brain tumor undergoing radiation therapy (RT). The primary results showed that neuroticism was associated with low physical well-being and psychoticism was associated with improved physical fitness scores after RT. The authors suggest these personality traits should be considered in personalized RT plans. The authors are commended on their use of objective measures of physical performance and ability to recruit a small sample despite the unique challenges in recruiting brain tumor patients actively in treatment. A gap in knowledge on this topic was clearly identified and described. While the study design has strengths, it likely does not close the gap in knowledge, but may make a modest contribution to the science due to several study limitations, noted below.

·       Please comment on the participant recruitment and retention procedures. It is commendable that the authors were able to able recruit 29 participants undergoing RT; however, it is unclear if the sample was sufficient enough to detect significant and clinically meaningful results. The study was likely underpowered due to the sample size, yet this was not discussed in the methods. The authors do mention this is a pilot study in the limitations section, but no where else in the manuscript setting the expectation that it would be a fully powered study. This is especially important as the authors assert their results should inform clinical practice in the discussion/conclusion.

·       Could the authors control for any covariates? For instance, 41% of the sample also underwent tumor resection and 79% underwent chemotherapy. These treatments could have also affected physical and psychological functioning?

·       What were the author’s hypotheses at the outset of the study?

·       Data collection: how much time passed between assessments? Was change in self-report and objective measures expected immediately after RT?

·       The authors performed basic statistical analyses and might comment further on this with supporting rationale.

·       The first four paragraphs of the discussion were redundant with the introduction section.

·       The limitations section should specify generalizability limitations, especially give the small sample size.

·       Since these results are not generalizable, the conclusion section makes assertions beyond the scope of the results. Perhaps the authors could rephrase in less absolute terms.

·       It would be great for the authors to comment on future research directions.

Author Response

Dear Reviewer 2,

we would like to express our gratitude for the Reviewers’ invaluable comments and suggestions for the improvements to our manuscript. All comments have been addressed and all changes suggested by the Reviewer have been implemented. Our answers are listed below.

This study aimed to evaluate the psychological/personality factors associated with post-treatment physical status and quality of life (QOL) in patients with brain tumor undergoing radiation therapy (RT). The primary results showed that neuroticism was associated with low physical well-being and psychoticism was associated with improved physical fitness scores after RT. The authors suggest these personality traits should be considered in personalized RT plans. The authors are commended on their use of objective measures of physical performance and ability to recruit a small sample despite the unique challenges in recruiting brain tumor patients actively in treatment. A gap in knowledge on this topic was clearly identified and described. While the study design has strengths, it likely does not close the gap in knowledge, but may make a modest contribution to the science due to several study limitations, noted below.

Comment 1. Please comment on the participant recruitment and retention procedures. It is commendable that the authors were able to able recruit 29 participants undergoing RT; however, it is unclear if the sample was sufficient enough to detect significant and clinically meaningful results. The study was likely underpowered due to the sample size, yet this was not discussed in the methods. The authors do mention this is a pilot study in the limitations section, but no where else in the manuscript setting the expectation that it would be a fully powered study. This is especially important as the authors assert their results should inform clinical practice in the discussion/conclusion.

Thank you very much for your comment. We agree that the sample size used in our study is not sufficient to generalize the results to the entire population. However, we noted that the study is a pilot study, which we additionally emphasized in the material and methods section. Earlier studies in the group of patients with brain tumors undergoing radiation therapy have similar sample sizes. We present a few of them below.

Kepka L, Tyc-Szczepaniak D, Osowiecka K, Sprawka A, TrÄ…bska-Kluch B, Czeremszynska B. Quality of life after whole brain radiotherapy compared with radiosurgery of the tumor bed: results from a randomized trial. Clin Transl Oncol. 2018;20(2):150-159. doi:10.1007/s12094-017-1703-5

Bitterlich C, Vordermark D. Analysis of health-related quality of life in patients with brain tumors prior and subsequent to radiotherapy. Oncol Lett. 2017;14(2):1841-1846. doi:10.3892/ol.2017.6310

Kim KS, Wee CW, Seok JY, et al. Hippocampus-sparing radiotherapy using volumetric modulated arc therapy (VMAT) to the primary brain tumor: the result of dosimetric study and neurocognitive function assessment. Radiat Oncol. 2018;13(1):29. doi:10.1186/s13014-018-0975-4

We recognize that further multicentre studies, in larger groups and with longer follow-up times, are needed to generate more significant data. However, we are of the opinion that our pilot study will encourage other researchers to undertake further research on this important subject regarding personalized radiotherapy.

In addition, the sample size was determined for the correlation test, with the assumed test power of 80% and p = 0.05. The effect size was established based on previous publications on the relationship of personality in other types of cancer and was set at 0.5. For such assumptions, the minimum sample size is 28 participants.

As recommended, we have redrafted our final conclusions.

Comment 2. Could the authors control for any covariates? For instance, 41% of the sample also underwent tumor resection and 79% underwent chemotherapy. These treatments could have also affected physical and psychological functioning?

Thank you for your important comment. Considering a small sample size, we decided not to make additional subgroups and not to use statistical modeling that would allow for the control of confounding variables. In future studies with a larger group, we plan to use a multiple regression model to analyze the results, as noted by the reviewer.

Comment 3. What were the author’s hypotheses at the outset of the study?

Thank you very much for the pertinent question. In our study, we decided to check whether personality traits can affect how patients tolerate RT in terms of physical fitness and QoL. We hypothesized that differences in personality could be used to personalize RT plans.

Comment 4. Data collection: how much time passed between assessments? Was change in self-report and objective measures expected immediately after RT?

Thank you very much for the right question. Six weeks (30 days of irradiation, 5 times a week) elapsed between the initial and final evaluation. To the best of our knowledge, acute complications of RT, including fatigue, can appear during radiotherapy.

Comment 5. The authors performed basic statistical analyses and might comment further on this with supporting rationale.

Thank you very much for your comment. The sample size was determined for the correlation test, with the assumed test power of 80% and p = 0.05. The effect size was established based on previous publications on the relationship of personality in other types of cancer and was set at 0.5. For such assumptions, the minimum sample size is 28 participants.

Comment 6. The first four paragraphs of the discussion were redundant with the introduction section.

We have made a few changes as recommended, thank you.

Comment 7. The limitations section should specify generalizability limitations, especially give the small sample size.

Thank you. The sentence has been heavily reedited according to your comments. We hope we have not missed anything.

Comment 8.  Since these results are not generalizable, the conclusion section makes assertions beyond the scope of the results. Perhaps the authors could rephrase in less absolute terms.

Thank you for your valuable comment. We redacted the conclusions as recommended.

Comment 9. It would be great for the authors to comment on future research directions.

Thank you for your valuable comment. We supplemented the manuscript with a discussion of future research directions in the conclusion section.

Round 2

Reviewer 1 Report

I think that the authors have politely responded and corrected their manuscript based on my suggestion.

Reviewer 2 Report

Previous concerns were addressed. The authors might consider stating this is a pilot study in the introduction so it is clear from the outset.